# The Lifetime Expenditure in People with Keratoconus in Saudi Arabia

Saleha Al-Atawi [1,*], Ali Alghamdi [2] and Khaled Alzahrani [3]

[1] Optometry Department, Faculty of Applied Medical Science, Al Baha University, Al Baha 4781, Saudi Arabia
[2] Faculty of Medicine, Al Baha University, Al Baha 4781, Saudi Arabia
[3] Optometry Department, King Fahad Armed Hospital, Jeddah 23311, Saudi Arabia
*   Correspondence: sabufrakah@bu.edu.sa; Tel.: +966-17-7257700

**Abstract:** Aim: This study measured and evaluated the socioeconomic burden of people living with keratoconus in Saudi Arabia. Methods: This study employed a cross-sectional design, a Keratoconus Economic Burden Questionnaire, and a convenient sample of 89 keratoconus patients (58.4% male) drawn from multiple regions in Saudi Arabia. It was conducted using online surveys, and the data were analyzed using appropriate quantitative techniques. Results: The mean age and annual income of the participants were 33.24 (SD = 6.82) years and USD 9046.52 (SD = 16,866.48), respectively, with only 37% being employed for wages. Up to 94.4% needed glasses or contact lenses at least once a week, and 73.0% received care from optometrists. The condition forced 45.9% of the respondents to change careers or leisure activities, with a further 51.3% having to take time off work. The mean annual out-of-pocket expenses for buying and maintaining glasses or contact lenses, as well as traveling and accommodation for keratoconus-related treatment were USD 2341.76 (SD = 3053.09), with 48.32 incurring upwards of USD 3240 over the period. The treatment costs increased with disease duration, $r(89) = 0.216$, $p < 0.05$. Regression results showed that the existence of comorbid eye disease, changing glasses at least once a year, and wearing either glasses or contact lenses at least once a week individually had statistically significant negative effects on the total annual keratoconus treatment costs, while disease duration, utilization of optometrists, and taking time off had a statistically significant increase on the total cost ($p < 0.05$). Conclusion: With a prevalence rate of 1 in 375, progressive debilitation, and the lifetime nature of the disease, keratoconus is a critical public health concern in Saudi Arabia. The resulting visual impairment and discomfort, as well as both direct and indirect economic burdens, have considerable impacts on the patient's quality of life.

**Keywords:** keratoconus; lifetime expenditure; economic burden; keratoconus economic burden questionnaire

## 1. Introduction

Keratoconus is an ectatic disorder characterized by progressive thinning, and protrusion of the cornea, which results in irregular astigmatism and impaired vision. In advanced cases, scarring and opacity may occur, and corneal transplantation can be required [1–3]. In the early stages, keratoconus may be managed using standard spectacles or contact lenses, even though worsening astigmatism necessitates scleral or contact lenses. Highly progressive cases render it difficult to fit rigid contact lenses, by which time corneal transplantation becomes necessary [4]. Other than INTACS® (Toronto, ON, Canada), penetrating keratoplasty is easily one of the most common surgical approaches, even though the latter procedure is fraught with fairly high rates of post-operative complications, including secondary glaucoma, cataracts, and graft rejection [4–7]. UV light-mediated corneal collagen crosslinking has been used for the last two decades and is the only treatment that has been shown to be effective in slowing the progression of the disease [1,8–11].

While the exact etiopathogenesis of KC remains elusive, both hereditary and environmental factors have been associated with the condition, and although for many years it

was considered a non-inflammatory condition, for a couple of decades, a growing body of evidence has been accumulating that suggests the participation of inflammatory pathways in the onset and progression of the disease. The prevalence of KC is highly variable in various geographical areas, and with the greater availability of corneal topography devices in recent decades, diagnostic sensitivity has increased. Classically, based on a North American study, the prevalence was considered to be about 1 in 2000, but rates as high as 1 in 25 people have been reported in Southeast Asian populations [2,4,12–14]. In a study of the mandatory health insurance records of 4.4 million patients aged 10–40 in the Netherlands, for example, Godefrooij et al. [4] estimated the prevalence of keratoconus at 1 in 375 (95% Confidence Interval (CI)).

Usually, the normal cornea is prolate, i.e., more curve in the center, and its main meridians are orthogonal. In patients with keratoconus, the corneal apex bulges eccentrically more frequently in the inferotemporal quadrant, causing irregular astigmatism [2]. As the disorder is usually slowly progressive, the corneal shape and extent of astigmatism are typically mild at onset, which is why early-stage keratoconus is correctable with either glasses or soft contact lenses [1,2,15]. Rigid gas-permeable hard contact lenses, including special aspheric and multi-curve designs (which are preferable in advanced cases with a more uneven corneal surface), are the most frequent non-surgical treatment alternative currently used worldwide for patients with clinically significant keratoconus [7,8]. However, whether rigid gas-permeable contact lenses can hasten the progression of the disease is an unsettled topic, and further studies are warranted. Scleral lenses, which vault over the limbus and cornea without having contact with them, on the other hand, are more expensive and difficult to fit, but they can provide good vision even in very irregular corneas, and their use is increasing [2,8,15–17]. Eventually, however, a percentage of eyes with advanced keratoconus and intolerance to contact lenses or poor visual acuity, even with these optical devices will require a corneal transplant [1,2,9].

## 2. Problem Statement

Empirical evidence shows that the prevalence of keratoconus in Saudi Arabia is comparably higher than in other countries, possibly because of geographical/regional, environmental, and genetic differences [3,5,13,14,18]. A 2018 pediatric survey involving 522 patients (aged 6 to 21) estimated keratoconus prevalence to be 4.79% (95% CI = 2.96–6.62) [13]. Two studies among individuals seeking refractive surgery have shown a prevalence of manifest keratoconus between 8.6% and 24% in Saudi Arabia. Although due to selection bias, these studies could show a higher prevalence than in the general population, it is possible to compare them with similar studies in other countries, and these rates were higher than those found in refractive surgery services in Central Europe, North America, and South America [18,19]. Given the high incidence of keratoconus in Saudi Arabia [14,18,19], at least from the available empirical evidence and the scarcity of research on the disorder's socioeconomic burden [1,4], the proposed study sought to estimate its economic effects on patients in the Kingdom of Saudi Arabia.

### 2.1. Aim

To measure and evaluate the socioeconomic burden of people living with keratoconus.

Objectives

i.      To estimate keratoconus-related lifetime expenditure in Saudi Arabia;
ii.     To evaluate the socioeconomic burden on people with keratoconus and medical insurance;
iii.    To provide recommendations to overcome the economic burden on patients in Saudi Arabia.

## 3. Methodology

### 3.1. Time Horizon

This study relied on a cross-sectional design. A longitudinal design is not only resource- and time-intensive; it is unlikely to yield a comparably richer dataset than a cross-sectional design.

### 3.2. Sampling

The sample was drawn from various regions in Saudi Arabia. It comprised all people who had been diagnosed with keratoconus in one or both eyes and were asked to participate in the study. The study used convenience sampling. The participants were recruited both directly and through optometric and ophthalmology clinics in the Kingdom of Saudi Arabia. Respondents who suffered from acute, chronic, or genetic/congenital non-visual comorbid conditions, such as Downs Syndrome, Marfan syndrome, GAPO syndrome, osteogenesis imperfecta, and Ehlers–Danlos syndrome, were excluded [20].

### 3.3. Data Collection and Analysis Methods

Demographic and clinical history data were collected by way of structured questionnaires. Data on the effects of keratoconus on expenditures (including treatment and travel) were gathered using a keratoconus health expenditure checklist [1]. The questionnaires were administered in the form of online surveys (see Appendices B and C). Appropriate descriptive and inferential statistics were computed using SPSS, Eviews, and Microsoft Excel.

### 3.4. Validity and Reliability

To ensure construct validity and reliability, the researcher developed data collection tools through a review of the extant empirical and theoretical literature. The resulting tools were piloted using a jury of two experts in the field of optometry and ophthalmology. The findings from the pilot study informed modifications to the tools to ensure they measure the required constructs accurately and reliably, as well as ensure that they can be reliably and efficiently administered. The reliability of the tools was evaluated by way of the Cronbach's alpha test.

### 3.5. Ethical Considerations

The researcher sought approval from the Institutional Review Board at Al Baha University (protocol code 1443-21-43110073 and date of approval 21 March 2022). Standard safeguards, including informed consent, participant anonymization, transparency, integrity, confidentiality, and physical/digital security were strictly observed [21].

## 4. Results

### 4.1. Demographics

Data were gathered between March and June 2022. A total of 89 complete questionnaires from participants who had all been diagnosed with keratoconus were received at the end of the data collection period. The average age of the respondents was 33.24 years (Standard Deviation (SD) = 6.80). The average annual income was USD 9046.52 (SD = 16,798.42). Up to 46.07% of the respondents indicated that they had no income over the preceding 12 months. Table 1 summarizes the participants' demographic attributes.

All respondents were diagnosed with keratoconus before their 20th birthday, with 55.1% and 41.6% of the respondents diagnosed with keratoconus aged 10–14 years and 15–19 years, respectively. Accordingly, it had been 5–14 years since 73% of the respondents were diagnosed with the condition. As many as 80.9% had keratoconus in both eyes, while 7.9% and 11.2% had keratoconus in the left and right eyes, respectively. Up to 12.4% of the respondents had comorbid conditions, including dryness, cataracts, and allergies. While 33% of the respondents did not buy any glasses at all over the preceding 12 months, 25.8% and 22.5% bought glasses once and twice over the same period, respectively. At least 14%

reported buying glasses more than three times over the previous 12 months. Half of the respondents were either employed or self-employed, with as many as 9.5% reporting that they were unemployed due to keratoconus. See Figure 1 below.

**Table 1.** Participants' demographics.

| Demographics | Category | No. of Respondents | Percent |
|---|---|---|---|
| Gender | Female | 37 | 41.6% |
| | Male | 52 | 58.4% |
| Age | Less than 24 years | 12 | 13.5% |
| | 25–29 years | 12 | 13.5% |
| | 30–34 years | 28 | 31.5% |
| | 35–39 years | 26 | 29.2% |
| | 40–44 years | 6 | 6.7% |
| | 45–49 years | 4 | 4.5% |
| | Above 50 years | 1 | 1.1% |
| Annual income (USD) | 0 | 41 | 46.07% |
| | 2500 | 8 | 8.99% |
| | 5000 | 14 | 15.73% |
| | 15,000 | 8 | 8.99% |
| | 35,000 | 11 | 12.36% |
| | 50,000 | 5 | 5.62% |
| | More | 2 | 2.25% |

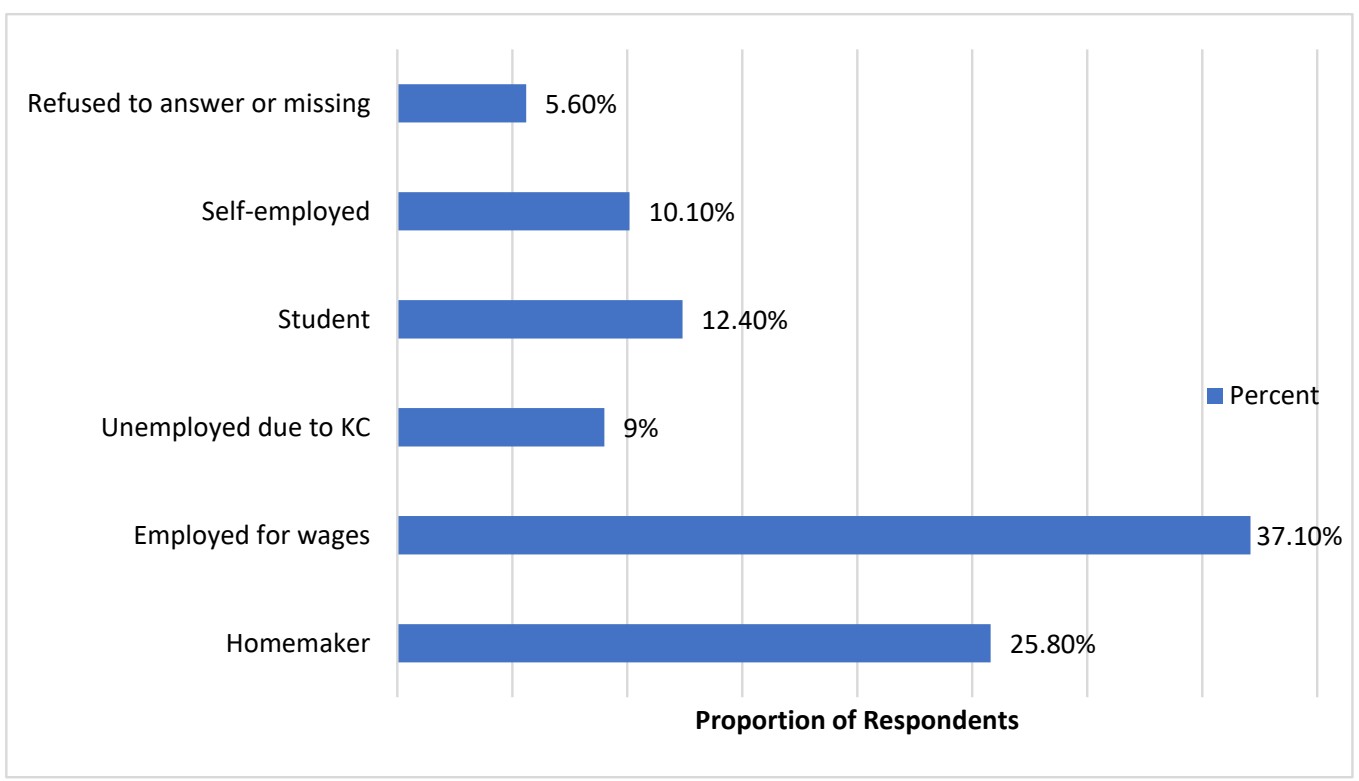

**Figure 1.** Participants' occupations.

Up to 42.7% of the respondents reported not having received surgical treatment for keratoconus, while 28.1% had undergone corneal transplantation. A further 23.6% and 5.6% wear scleral lenses and INTACS®, respectively. See Figure 2.

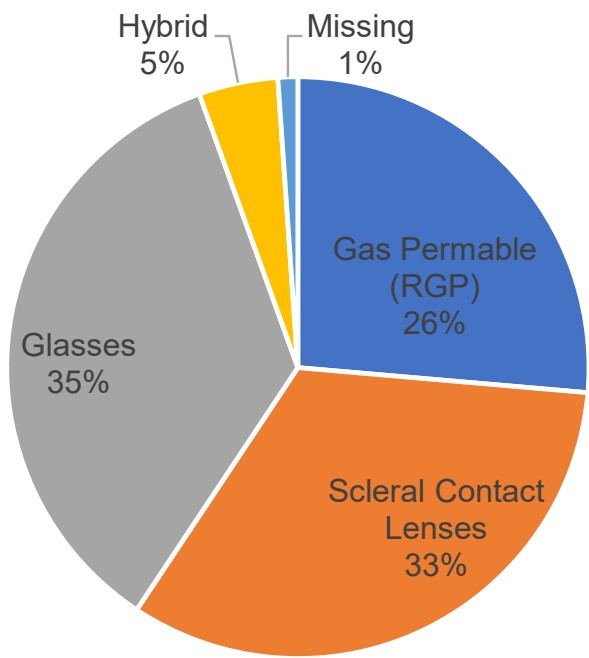

**Figure 2.** Keratoconus treatments received.

The results similarly showed that 36.0%, 33.7%, and 24.7% of the respondents used glasses, scleral lenses, and rigid gas permeable lenses, respectively, while 4.5% used a hybrid of technologies. Of those who used glasses, contact lenses, or other assistive technologies, 56% reported normal visual acuity in either eye, while 18.2% had normal visual acuity in both eyes. At least 26% reported not knowing their visual acuity. See Figure 3.

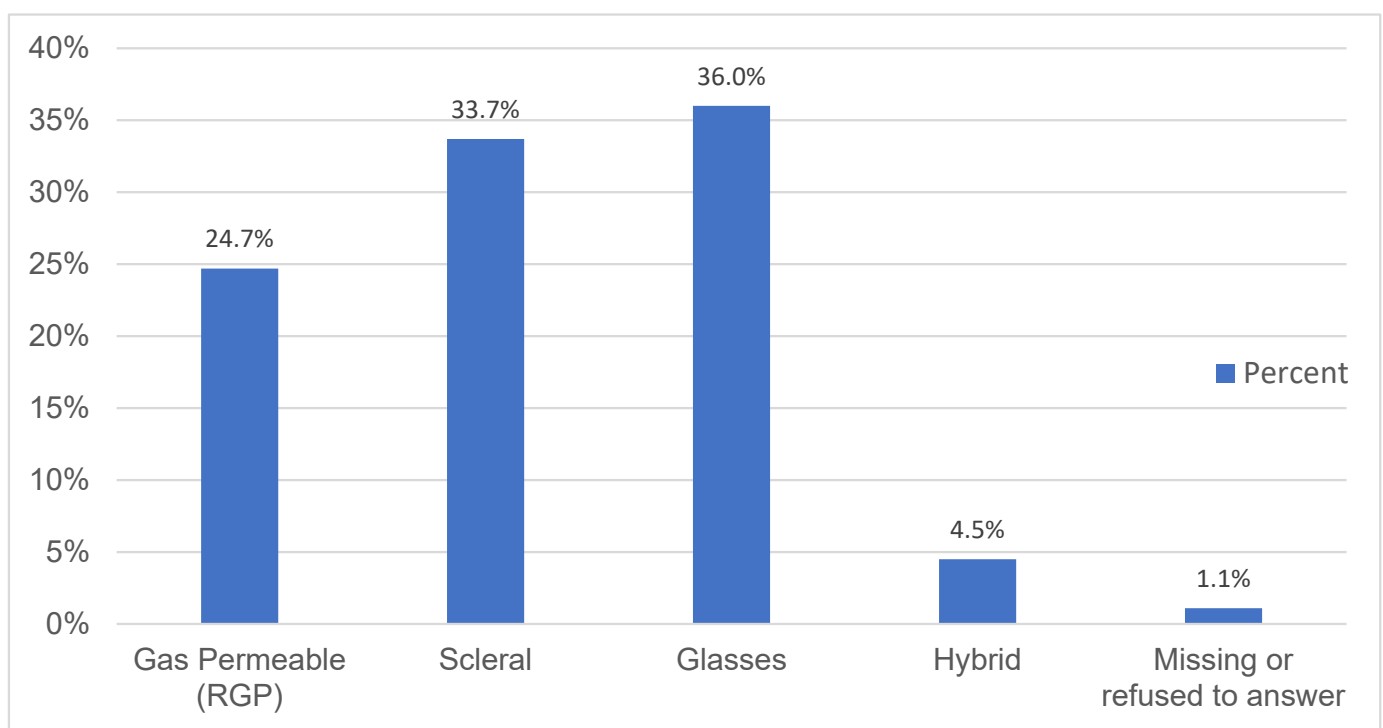

**Figure 3.** Assistive technologies.

Only 5.6% reported never needing glasses or other assistive technologies to see well, with 11.5% and 82.8% reporting needing glasses once or twice a week and many times a week, respectively.

### 4.2. KC-Related Disability and Productivity Losses

The results indicated that 45.9% of the respondents were forced to change careers, jobs, leisure activities, and/or courses of study on account of keratoconus. A further 51.3% reported having had to take time off work or having been indisposed to work either because of their condition or the need to receive treatment/care for keratoconus. The reasons for these changes included occupational disability (e.g., inability to cope with dusty work environments, failing mandated medical exams, and occupations or hobbies that require excellent vision), the necessity to seek adequate treatment, and prolonged symptoms/discomfort (headaches, blurred vision, deterioration of vision in the day, and eye strain). See Appendix A. The resulting disability was such that 47% of the respondents were unable, at least once over the preceding twelve months, to care for themselves.

### 4.3. Types of Care

The majority of the respondents attended private clinics for keratoconus care (52.8%). Up to 73.0% of the respondents reported receiving care from optometrists, while 27% did not. Most of the services sought by non-optometrists included designing and fitting lenses. Figure 4 summarizes the results.

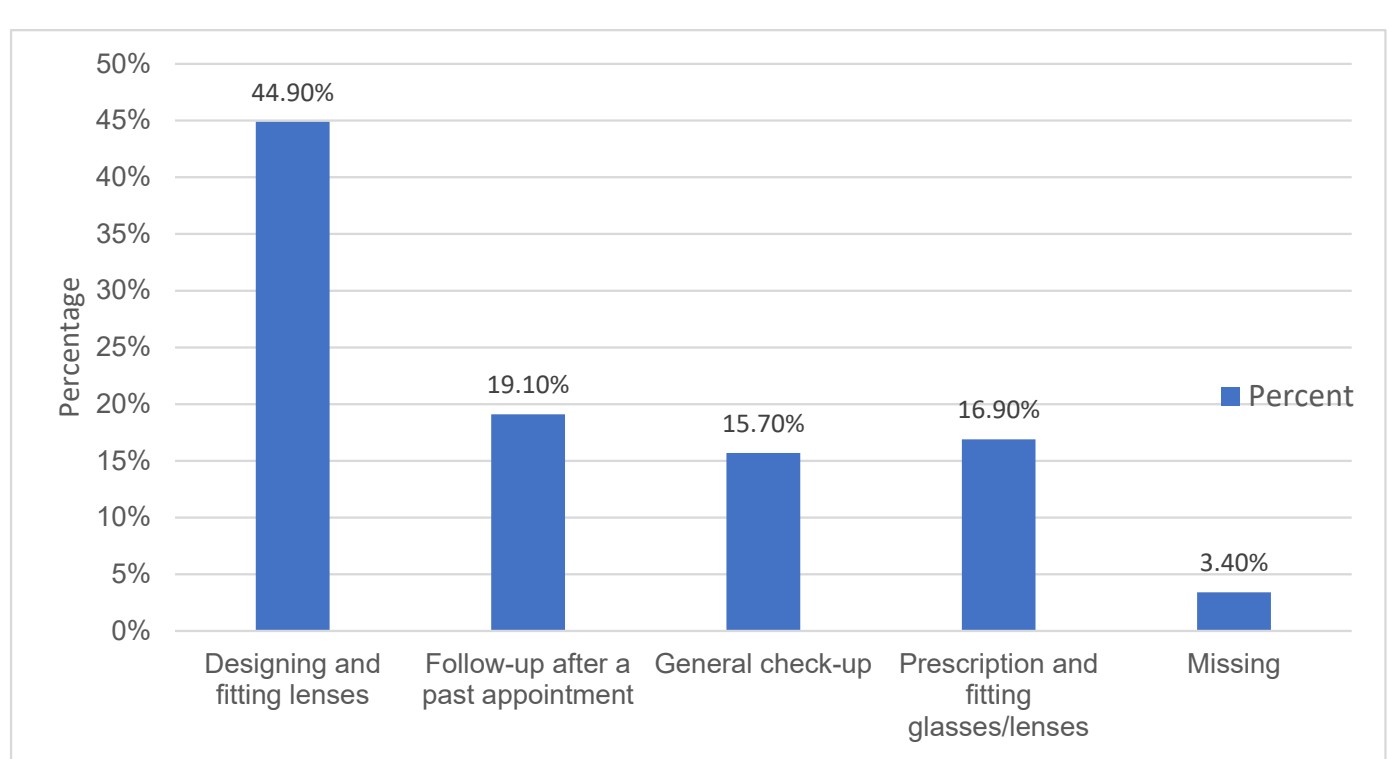

**Figure 4.** Optometric services sought.

At least 38.2% of the respondents sought care from other specialists, practitioners, and/or hospitals, either in addition to or instead of optometrists. The services sought were nearly identical to those sought by optometrists. See Figure 5.

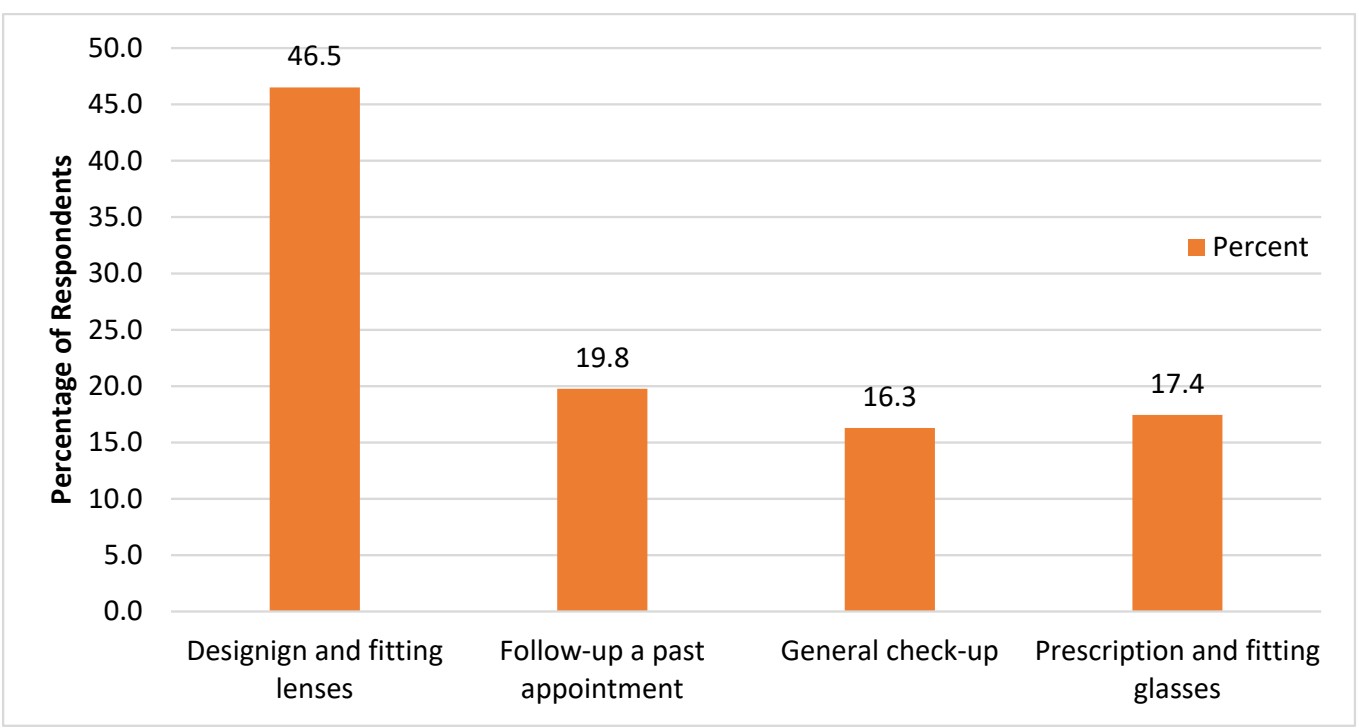

**Figure 5.** Non-optometric services.

Only 37.1% of the respondents did not buy glasses in the preceding twelve months, with 25.8%, 22.5%, and 14.6% of the respondents reporting having bought glasses once, twice, and more than thrice in the past twelve months.

### 4.4. Treatment Expenditure

The mean out-of-pocket expenses and other expenses over the preceding year amounted to USD 2341.76 (SD = 2826.82). See Table 2.

**Table 2.** Out of pocket expenses.

| Expenses | Min. | Max. | Mean | Std. Deviation |
|---|---|---|---|---|
| How much out-of-pocket expenses did you incur to buy glasses or contact lenses over the last 12 months? | 0 | 2025 | 780.62 | 566.21 |
| How much money do you spend to take care of your contact lenses and glasses (e.g., wipes)? | 0 | 2700.00 | 382.17 | 506.85 |
| How much, in out-of-pocket expenses, did you incur on transport and accommodation related to keratoconus surgery? | 0 | 22,950.00 | 1155.88 | 2645.47 |

More than 30% of the respondents spent USD 5500 or more, while 17.98% and 15.73% incurred USD 2500 and USD 3500, respectively. See Table 3 below.

On average, the cost of buying glasses and surgery-related transport/accommodation costs related to keratoconus surgery accounted for the largest cost drivers for the majority of the respondents. Buying glasses or contact lenses cost USD 780.622 (SD = 566.21), on average, with about a quarter of all respondents spending USD 1500 or more. See Table 4 below.

Additionally, Table 5 shows that respondents spent an average of USD 382.17 (SD = 506.85) on supplies to maintain glasses, contact lenses, and other technologies. About 17% and 30% of the respondents spent in excess of USD 1000 and 500, respectively.

**Table 3.** Total KC-related medical expenses.

| Total KC-Related Medical Costs (USD) | No. of Respondents | Percentage |
|---|---|---|
| 0 | 2 | 2.25% |
| 500 | 9 | 10.11% |
| 1000 | 14 | 15.73% |
| 1500 | 7 | 7.87% |
| 2500 | 16 | 17.98% |
| 3500 | 14 | 15.73% |
| 5500 | 12 | 13.48% |
| More | 15 | 16.85% |

**Table 4.** Out-of-pocket expenses incurred to buy glasses or contact lenses over the last 12 months (USD).

| How Much Out-of-Pocket Expenses Did You Incur to Buy Glasses or Contact Lenses over the Last 12 Months (USD)? | No. of Respondents | Percentage |
|---|---|---|
| 0 | 4 | 4.49% |
| 250 | 11 | 12.36% |
| 500 | 20 | 22.47% |
| 750 | 8 | 8.99% |
| 1000 | 19 | 21.35% |
| 1250 | 4 | 4.49% |
| 1500 | 12 | 13.48% |
| 2000 | 7 | 7.87% |
| More | 4 | 4.49% |

**Table 5.** Expenses on caring for contact lenses and glasses.

| How Much Money Do You Spend to Take Care of Your Contact Lenses and Glasses (e.g., Wipes)? | Frequency | Percent |
|---|---|---|
| 0 | 12 | 13.48% |
| 125 | 20 | 22.47% |
| 250 | 14 | 15.73% |
| 500 | 21 | 23.60% |
| 750 | 6 | 6.74% |
| 1000 | 8 | 8.99% |
| 1500 | 3 | 3.37% |
| More | 5 | 5.62% |

While 31.46% of the respondents reported having spent nothing over the twelve months, 16.85%, 15.75%, 10.11%, and 11.24% spent USD 500, 1000, 1500, and 2000, respectively. About a tenth of the respondents spent more than USD 2500. See Table 6.

**Table 6.** Out-of-pocket expenses incurred on treatment over the past 12 months.

| How Much, in Out-of-Pocket Expenses, Did You Incur on Your Treatment over the Past 12 Months (USD)? | No. of Respondents | Percent |
|---|---|---|
| 0 | 28 | 31.46% |
| 500 | 15 | 16.85% |
| 1000 | 14 | 15.73% |
| 1500 | 9 | 10.11% |
| 2000 | 10 | 11.24% |
| 2500 | 6 | 6.74% |
| 3000 | 4 | 4.49% |
| More | 3 | 3.37% |

As shown in Figure 6, the out-of-pocket transport and accommodation costs related to keratoconus surgery averaged USD 1155.88 (SD = 2645.47). While 30% of the respondents

did not incur surgery-related travel and accommodation costs, 16.5%, 15.4%, 9.9%, and 11.0% reported having spent USD 500, 1000, 1500, and 2000, respectively. Approximately 13% of the respondents incurred more than USD 2500 in transport costs in visits to clinics for keratoconus care over one year.

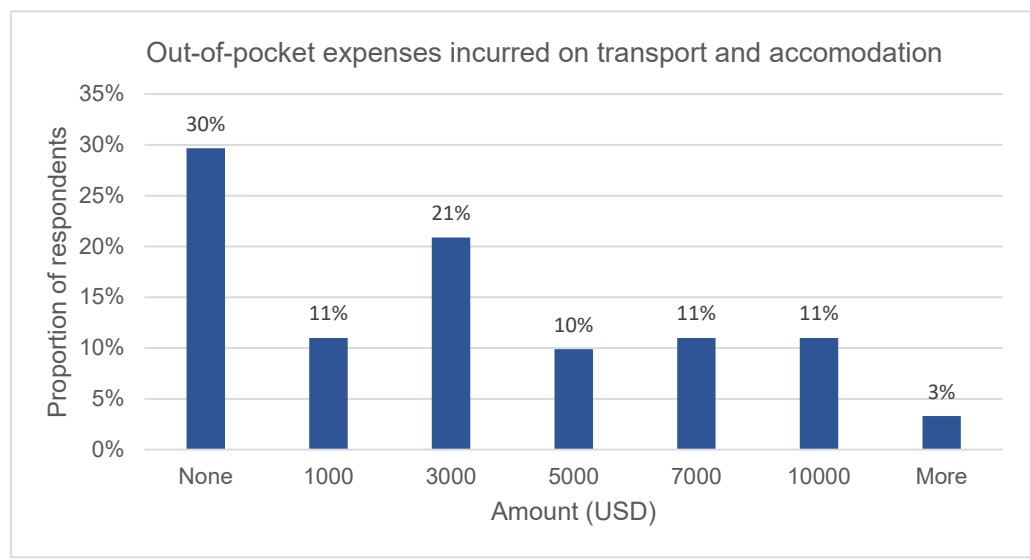

**Figure 6.** Non-surgery-related costs.

Other than the costs of buying glasses, lenses, and supplies, the respondents incurred more costs, including costs for consultations, check-ups, testing, lens fitting, hospitalization, and surgical fees. The majority of the respondents spent less than USD 675 over one year. See Figure 7 below.

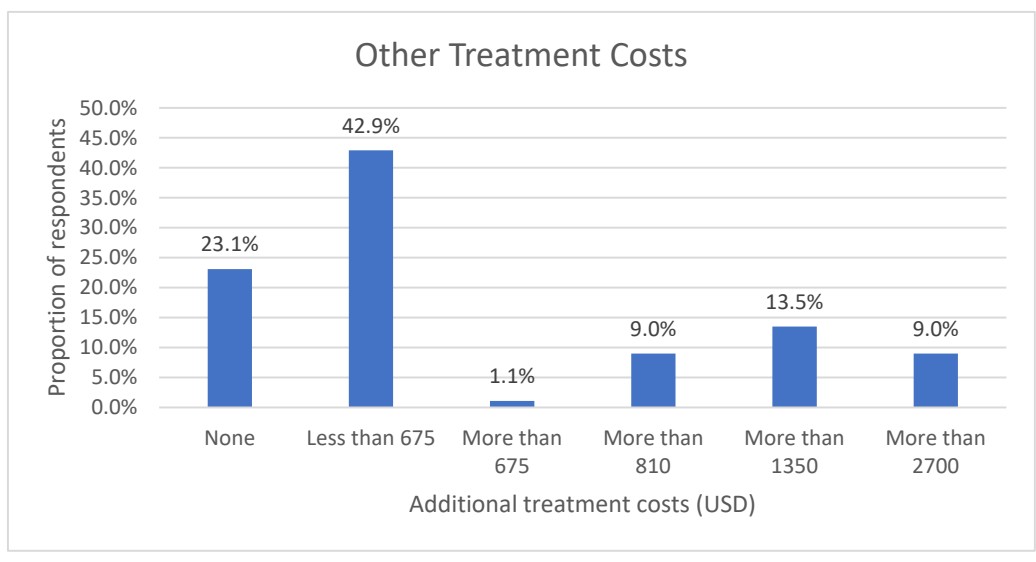

**Figure 7.** Additional treatment costs.

Overall, the line of best fit in Figure 8 shows that annual expenditure tended to increase with the number of years living with keratoconus. Even so, the linear correlation coefficient was 0.038, which was a near negligible co-variation of the duration since diagnosis and the twelve-month expenditure. This outcome could be explained by the fact that a twelve-month period of observation was relatively short since major treatment expenditures (such as surgery and change of glasses) could remain relatively constant within a given year.

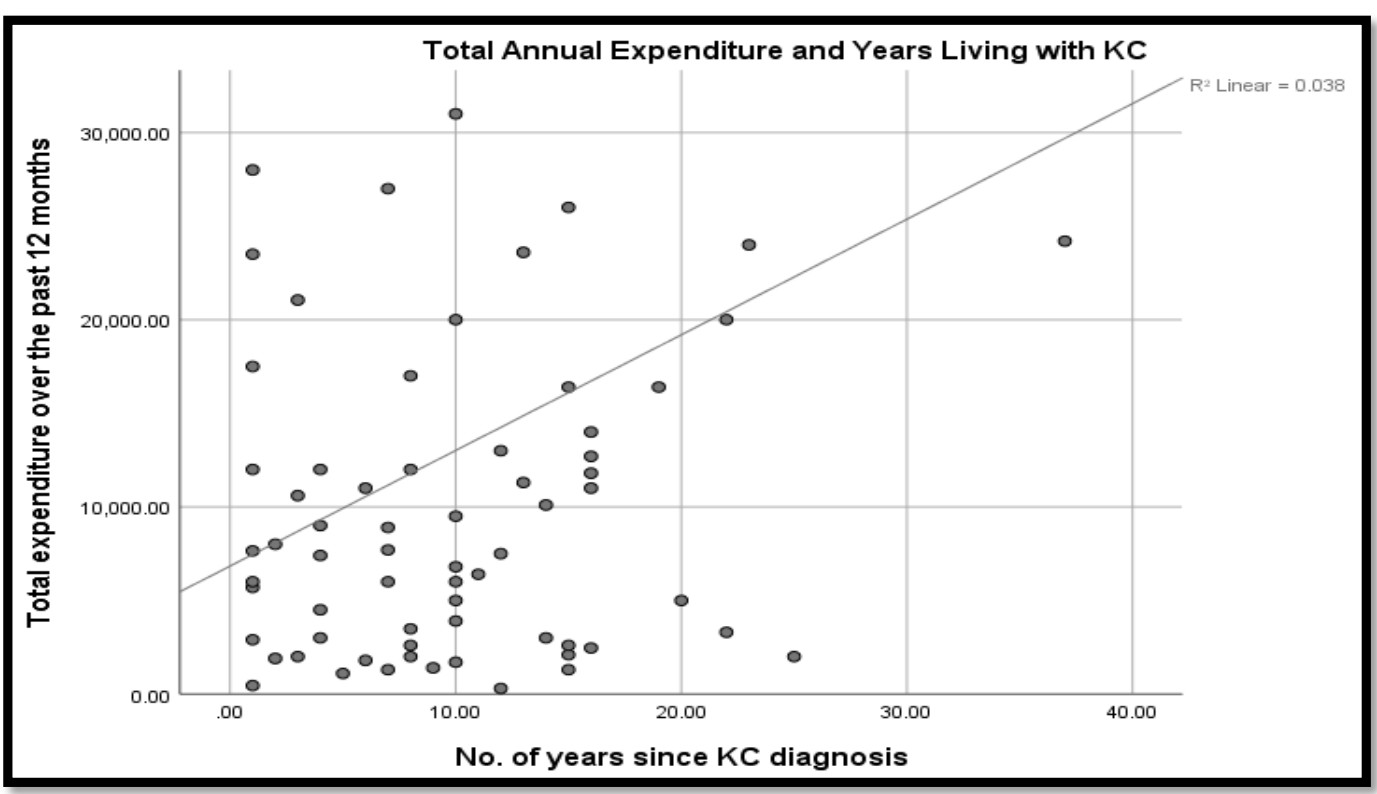

**Figure 8.** Annual expenditure increases with the number of years with KC.

Spearman's rho test showed the coefficient of correlation between the duration of disease and the total expenditure was positive and statistically significant, $r(89) = 0.216$, $p < 0.05$. Similarly, there were multiple, statistically significant intercorrelations between multiple possible cost predictor variables, at a five percent significance level. Notably, however, the coefficient of correlation of the frequency with which patients needed to wear glasses or contact lenses to see well and the existence of comorbid eye disease was negatively and statistically significant at 1 percent. The correlation coefficient summary is given in Table 7.

*4.5. Insurance Coverage*

Table 8 shows that at least 42% of the keratoconus patients sampled did not have private insurance. Only 7.9% of the respondents indicated that keratoconus was covered by their private insurance policy. Up to 87.6% did not individually pay for insurance premiums. The rest of the respondents paid premiums for private insurance premiums, which ranged between USD 945 and 2700 for the majority of them. See Table 8 below.

More than 60% and 24% of the respondents believed that private insurance premiums were inaccessible, and the coverage offered poor value for money. At least 9% of the respondents believed private insurance coverage was unnecessary. Of the 48.3% who had private insurance, 92.96% were dissatisfied with the insurance rebates that they received to cover keratoconus treatment and other related care. They believed the rebates needed to cover more treatment and care expenses, including glasses, eye drops, and surgical expenses. See Figure 9 below.

The correlation analysis did not, however, indicate causation. To ascertain whether the total cost could be predicted by any of the variables, linear regression models were developed. The resulting variables that predicted the total cost are shown in Table 9.

**Table 7.** Correlation analysis.

| Column Title | Years Since Diagnosis | Total Cost | Comorbid Conditions | Assistive Technology | Visual Acuity | Frequency of Buying Glasses | Frequency of Wearing Glasses | Treatment | Needed Carer | Annual Income | Age | Age of Diagnosis | Time Off Work |
|---|---|---|---|---|---|---|---|---|---|---|---|---|---|
| Years Since Diagnosis | 1.000 | 0.224 * | −0.128 | −0.174 | 0.060 | 0.001 | −0.007 | 0.286 ** | 0.100 | 0.097 | 0.405 ** | −0.271 * | 0.154 |
| Total Cost | 0.224 * | 1.000 | −0.027 | −0.300 ** | 0.075 | 0.035 | −0.040 | −0.029 | 0.332 ** | −0.053 | 0.214 * | −0.025 | 0.353 ** |
| Comorbid Conditions | −0.128 | −0.027 | 1.000 | 0.187 | 0.025 | −0.164 | −0.303 ** | 0.057 | 0.114 | 0.043 | −0.024 | 0.003 | 0.033 |
| Assistive Technology | −0.174 | −0.300 ** | 0.187 | 1.000 | −0.002 | 0.222 * | −0.094 | 0.153 | −0.030 | −0.031 | −0.102 | −0.049 | −0.029 |
| Visual Acuity | 0.060 | 0.075 | 0.025 | −0.002 | 1.000 | −0.089 | −0.199 | 0.096 | −0.033 | 0.177 | −0.093 | 0.081 | 0.186 |
| Frequency of wearing glasses | 0.001 | 0.035 | −0.164 | 0.222 * | −0.089 | 1.000 | 0.199 | 0.103 | −0.061 | 0.013 | 0.142 | −0.110 | 0.000 |
| Frequency of wearing glasses | −0.007 | −0.040 | −0.303 ** | −0.094 | −0.199 | 0.199 | 1.000 | −0.019 | 0.131 | 0.155 | 0.078 | 0.204 | 0.032 |
| Treatment | 0.286 ** | −0.029 | 0.057 | 0.153 | 0.096 | 0.103 | −0.019 | 1.000 | 0.122 | 0.014 | −0.043 | −0.173 | 0.351 ** |
| Needed Carer | 0.100 | 0.332 ** | 0.114 | −0.030 | −0.033 | −0.061 | 0.131 | 0.122 | 1.000 | −0.013 | 0.143 | −0.201 | 0.295 * |
| Annual Income | 0.097 | −0.053 | 0.043 | −0.031 | 0.177 | 0.013 | 0.155 | 0.014 | −0.013 | 1.000 | 0.127 | 0.122 | 0.152 |
| Age | 0.405 ** | 0.214 * | −0.024 | −0.102 | −0.093 | 0.142 | 0.078 | −0.043 | 0.143 | 0.127 | 1.000 | −0.062 | 0.162 |
| Age of Diagnosis | −0.271 * | −0.025 | 0.003 | −0.049 | 0.081 | −0.110 | 0.204 | −0.173 | −0.201 | 0.122 | −0.062 | 1.000 | −0.039 |
| Time off Work | 0.154 | 0.353 ** | 0.033 | −0.029 | 0.186 | 0.000 | 0.032 | 0.351 ** | 0.295 * | 0.152 | 0.162 | −0.039 | 1.000 |

* Correlation is significant at the 0.05 level (2-tailed). ** Correlation is significant at the 0.01 level (2-tailed).

**Table 8.** Private insurance premiums.

| How Much Money Do You Pay as Premiums for Your Private Insurance Cover per Year (USD)? | No. of Respondents | Percent | Cumulative Percent |
|---|---|---|---|
| Nothing | 78 | 87.6 | 87.6 |
| 945–1349 * | 7 | 7.9 | 95.5 |
| 1350–2699 | 2 | 2.2 | 97.8 |
| More than $2700 | 2 | 2.2 | 100.0 |

\* The researcher's preliminary market survey indicated that the minimum premium was USD 945. A zero premium indicates that the respondent did not have private insurance cover.

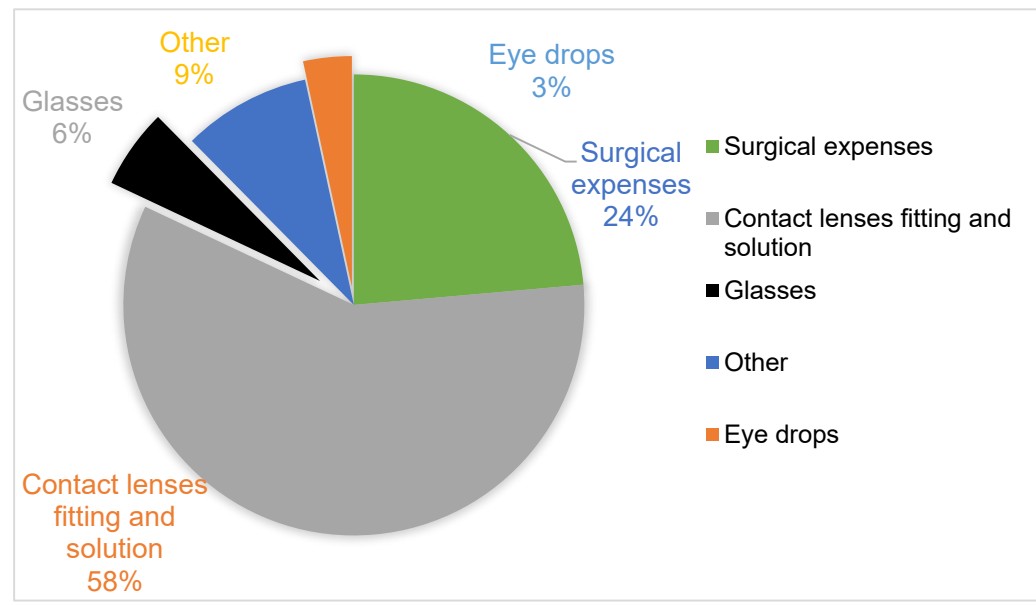

**Figure 9.** Expenses that should be included in insurance rebates.

**Table 9.** Regression analysis.

| Variable | Coefficient | Std. Error | *t*-Statistic | Probability. |
|---|---|---|---|---|
| Comorbid conditions | −3981.533 | 1479.328 | −2.691448 | 0.0087 |
| Frequency of buying glasses | −1808.047 | 1027.379 | −1.759864 | 0.0825 |
| Frequency of wearing glasses or lenses | −4944.126 | 1723.959 | −2.867891 | 0.0053 |
| Disease duration | 3342.617 | 1345.000 | 2.485217 | 0.0151 |
| Optometric care | 5154.051 | 2503.182 | 2.058999 | 0.0429 |
| Non-optometrist care | 3882.983 | 2305.681 | 1.684094 | 0.0963 |
| Time of work | 5480.097 | 2359.953 | 2.322121 | 0.0229 |
| Career change | 2188.030 | 2439.605 | 0.896879 | 0.3726 |
| Surgery | −2227.092 | 2562.879 | −0.868981 | 0.3876 |
| Needed carer | 3369.768 | 2493.617 | 1.351358 | 0.1806 |
| Type of clinic | 2721.511 | 1882.408 | 1.445761 | 0.1524 |
| R-squared | 0.314568 | Mean dependent var | | 8872.575 |
| Adjusted R-squared | 0.224379 | S.D. dependent var | | 11,323.79 |
| S.E. of regression | 9972.787 | Akaike info criterion | | 21.37081 |
| Sum squared residuals | 7.570009 | Schwarz criterion | | 21.68259 |
| Log likelihood | −918.6301 | Hannan−Quinn criteria. | | 21.49635 |
| Durbin-Watson stat | 2.064649 | | | |

The model could predict 31.46% of the variations in the total costs incurred in treatment, care, and lifestyle costs incurred on account of keratoconus. The existence of comorbid eye conditions significantly predicted inverse total cost scores, b = −3981.533, $t(87) = −2.69$, $p < 0.01$. Similarly, patients who had to wear glasses or contact lenses

to see well were likely to have lower costs than those who did not, b = $-1808.047$, $t(87) = -1.76$, $p < 0.01$. Undergoing surgery to correct keratoconus also has a negative effect on the lifestyle and medical costs of keratoconus, but this effect was not statistically significant, b = $-2227.09$, $t(87) = -0.87$, $p > 0.05$. On the contrary, disease duration, receiving care from optometrists, and taking time off had a statistically significant increase in the total cost of lifestyle and healthcare costs to the keratoconus patients. Other variables, including the inability to care for oneself, forced career or leisure activity change, receiving care from non-optometrists, and attending either a public or private clinic had a positive effect on the total costs, but their effects were not statistically significant ($p > 0.05$).

## 5. Discussion

The income profile of the respondents showed that the sample mainly comprised a low-income population. Saudi Arabia's GDP per capita was estimated at USD 20,110 in 2022 [22], implying that only 36% of the sample that had more than USD 13,500 fell in the average income category, while those with less or without income were either in the lower income brackets or were dependents. An estimated 42% of the respondents reported never having received surgical treatment for keratoconus, which, given the fact that corneal cross-linking is arguably the most effective treatment for keratoconus [10,23,24], points to potentially poor access to the best care. This finding is consistent with past empirical evidence that cases of keratoconus in Saudi Arabia (and other countries in the region) are relatively more prevalent and advanced at the time of diagnosis than in other parts of the world [25].

While studies elsewhere in the world found broad variations in prevalence, they also thought that these were related to factors that include ethnicity, geography, diagnostic criteria, and methodological differences [14,26]. With respect to geographic factors, however, environmental factors, such as ultraviolent light exposure and altitude, may account for variations [12,26,27]. Generally, research shows a high keratoconus prevalence in Saudi Arabia, Israel, and India compared to regions in North America, Europe, and parts of Asia [18,25]. Against an estimated global prevalence of 1 in every 2000 people, for example, Assiri's study in Asiri province found that the prevalence of keratoconus has shown 20 per 100,000 population and high disease severity, with an advanced stage keratoconus mean age of 17.7 (SD = 3.6) years [25].

Furthermore, the treatments received by participants in this study had lower levels of effectiveness, particularly concerning the progression of the disease. The extant empirical evidence shows that INTACS® are ideally indicated for mild or moderate cases that are intolerant to contact lenses and have clear optical zones [6,14,14,28,29]. They may be an alternative to rehabilitative lamellar or penetrating keratoplasty, as well as for uncorrected acuity [27,30,31]. Keratoconus is the main indication for scleral contact lenses for enhanced comfort, lens centration, and intolerance to corneal gas-permeable lenses [14,30]. Empirical evidence shows it can prevent corneal transplantation in up to 80% of severe keratoconus cases, even with lamellar keratoplasty [14,30]. Keratoplasty is indicated in cases of corneal scarring and lamellar or full thickness [31].

While the cost, diagnosis, access, and availability of corneal donors remains an impediment to transplantation [7,32], the finding that up to 28% of the respondents had undergone corneal transplantations is encouraging but may be accounted for by sampling issues. Given the range of alternatives, differing effectiveness and indications, more research is needed to ascertain the effectiveness, access, and adverse effects of INTACS®, scleral lenses, and corneal transplants for treating keratoconus in Saudi Arabia against evidence-based indications, as a basis for enhanced efficiency and effectiveness in the management of the condition.

A higher-than-average proportion of the respondents in the present study attended optometric clinics, primarily for prescribing, designing, and fitting glasses or lenses, with about half as many seeking similar services from non-optometric practitioners/facilities. Surgery was not indicated as a reason for visits, although 28% of the respondents had

since undergone corneal transplantation. The proportion of those that had undergone surgery was lower than 48% in Chan et al., and it was unclear whether the surgery involved cross-linking procedures [1].

### 5.1. Keratoconus Disability and Productivity Losses

This study showed that keratoconus diagnosis occurred fairly early on in the lives of patients, and thus the lifetime costs of the condition were likely to be cumulatively high. The age of diagnosis is consistent with small sample studies [24,32] but much lower than the estimated age-specific incidence of 1:7500 (13.3/100,000) established in large sample studies [4]. Godefrooij et al. [4], for example, determined that the average age of diagnosis is 28.3 years, but it is likely that the age of diagnosis, as against the age of onset, depends on access to care [14,18,33,34]. Other than the age of diagnosis and the fact that keratoconus is not considered a disability in Saudi Arabia (other than in rare cases where patients' visual acuity is severely compromised), the results showed that severe symptoms of the condition and resulting occupational and social disability, as well as the financial consequences, are substantial. This may be exacerbated by the evidence of low awareness of keratoconus in the general population in Saudi Arabia, resulting in policy inaction, low health-seeking behavior and difficult social/work environments [3,5,13,33]. Al-Dairi et al., for example, found that the prevalence of depression in a sample of keratoconus patients in Saudi Arabia was 40.6% ($n = 134$; $p < 0.001$) and further that the use of corrective lenses in both eyes heightened the risk of depression even higher [33].

The findings showed that close to half of respondents were forced to take time off work or alter their career, leisure, educational, and even professional choices on account of keratoconus, which potentially implies suboptimal decision-making with equally suboptimal financial and economic implications. The results showed that an estimated 10% of the sampled population were completely incapable of working or finding work due to keratoconus. Godefrooij et al. estimated the twelve-month losses at AUD 500 for an Australian sample [4].

### 5.2. Keratoconus Expenditure

The calculated out-of-pocket costs for treating and managing keratoconus over twelve months, including the out-of-pocket expenses for glasses, contact lenses, and supplies, averaged USD 2341.76. Additionally, the majority of keratoconus patients incurred USD 675 to 3510 on transport, accommodation, and other ancillary expenses in seeking treatment. While this study did not verify the participants' incomes, if indeed 46% of those sampled had no income and 15.73% had an annual income of not more than USD 2341.76, keratoconus potentially has debilitating economic effects on the patients. Unlike Godefrooij et al. [4], this study's findings showed that expenditure was a positive function of disease duration, possibly because the costs depended on the quality of treatment/care and whether or not such treatments stemmed from the progression of the condition [4,23,35].

With glasses, the condition is correctable in the early stages, but the failure to treat the underlying causal factors often fails to stem its progression [29,30,35]. Corneal collagen cross-linking can stop keratoconus progression, but it is often not covered by insurers despite leading to lower costs in the long term [1]. In a study to ascertain the cost-effectiveness of corneal collagen cross-linking in the USA, Canada, and Western Europe, Leung et al. [9], Salmon et al. [10], and Lindstrom et al. [11] established that patients who underwent corneal cross-linking enhanced their quality of life, were less likely to require penetrating keratoplasty, and incurred lower lifetime costs or productivity losses. They spent 27.9 fewer years in advanced keratoconus stages [24]. In Lindstrom et al.'s study, the direct medical costs for patients who underwent corneal collagen cross-linking were USD 8677 lower, i.e., USD 30,994 compared to USD 39,671. The per capita lifetime productivity gains associated with corneal cross-linking were estimated at USD 43,759 [10,11,24].

Unlike Godefrooij et al. [4], Leung et al. [9], Rebenitsch et al. [24], and Pinto et al. [33] this present study did not estimate the lifetime costs of the disease but focused on the

individual cost drivers as predictors of the overall lifetime costs. This is arguably more practically relevant information for patients, practitioners, and policymakers. At a 5% significance level, the regression results indicate that comorbid eye conditions, changing glasses frequently, and wearing glasses or contact lenses frequently are likely to result in lower lifestyle and medical costs of keratoconus. There are two possible explanations for this counterintuitive finding. Past studies have shown that prescriptions to treat comorbid conditions and medication usage tend to be significantly higher among patients with some other eye conditions, such as dry eye disease. There is similarly a relationship between ocular comorbidities and systemic diseases, such as diabetes, with implications for effective and efficient detection and management [10,34].

With a third of the sampled population in this study not changing their glasses, it appears that the direct cost incurred in buying glasses is significantly lower than the indirect costs of either not wearing glasses or using poor glasses. Specifically, changing glasses once in twelve months and wearing either glasses or contact lenses at least once a week are likely to result in USD 939.46 and 2815.84 lower annual total costs, respectively. In contrast, a five-year disease duration is likely to result in USD 1450.54. Patients who attended optometric clinics at least once in twelve months are likely to have USD 2419.94 more in total costs. The results are inconclusive of the cost impact of undergoing keratoconus surgery, attending either public or private clinics, and assisted living due to keratoconus. Thus, more research with larger and more robust sampling is required to settle these findings. It is notable that none of the respondents indicated having undergone corneal cross-linking, which is the only treatment (approved by the US Food and Drug Administration) that has been shown to stem the progression of keratoconus. The effectiveness of surgery could be a factor in the results, as those with penetrating keratoplasty struggles with high rates of post-operative complications [4–7].

Given the high hospital utilization by keratoconus patients and the high cost of care, the lack of health insurance and/or government cover for treatment and other costs has immense implications [1,10]. This study found that 73% sought optometric services over the preceding year, and close to 50% sought services from other services, rates which are comparable to higher utilization rates elsewhere. Similarly, more research is required to investigate the impact of insurance cover on health service utilization and health outcomes, including the age of diagnosis, health-seeking behavior for patients with keratoconus, and the treatments open to them. While the actual costs are likely a function of income and lifestyle factors [1], this study's finding of comparatively higher average out-of-pocket expenditures relative to the less than USD 1350 paid by the majority of respondents in premiums shows a possible need for increased insurance coverage. This study identified an existing need for health insurance policies to cover fitting contact lenses and lens solutions, surgical expenses, and glasses.

### 5.3. Type of Care

Keratoconus requires multi-disciplinary management, including primary eye care practitioners, general practitioners, ophthalmologists, and optometrists [24]. The condition is difficult to detect at early stages, and it is usually possible to achieve good visual acuity with standard glasses, resulting in the unchecked progression of the disease. Studies into the sequence of events leading up to the keratoconus diagnosis show a lack of awareness among patients and the criticality of referrals from primary points of contact to optometrists, ophthalmologists, and other specialists [11,36].

Collaboration is, however, little known, and efforts are usually geared toward most prevalent eye diseases, age-related disorders, and primary care referral patterns [36]. Advanced stage keratoconus is difficult to correct, and it is a common indicator of corneal surgery [10,23]. An estimated 20% of keratoconus patients require corneal transplantation [23]. This study showed the acceptably high utilization of both optometrists and other facilities, but there is a case to be made for the services offered by non-optometrists to increase from 38%. This is not least because the services sought by both optometrists and

other practitioners appear to be the same when more differentiated services are possible. The potential for co-management and referral of cases across specialist/practitioner groups and from primary care to specialist care levels exists in the diagnosis and effective and efficient management of keratoconus [35,36].

## 6. Conclusions

An understanding of the financial burden of keratoconus in Saudi Arabia is important. The fact that a majority of the respondents in this study were diagnosed with keratoconus before their 20th birthday places a clear emphasis on the lifetime economic burden, particularly given the lack of private insurance coverage. With just 5.6% of the respondents in this study reporting not using any assistive technology, the next line of inquiry should be on how well the technologies being used by keratoconus patients in Saudi Arabia are properly indicated, given the severity of the symptoms and other clinical considerations, as well as the socioeconomic barriers to attaining evidence-based practice in respect to the same. Further research is similarly needed to ascertain the availability and cost of cross-linking and other treatments that can stop the progression of keratoconus [9,11], including their comparative pharmaco-economic impact [24] and the capacity of optometrists, hospitals, and other facilities to offer the same in Saudi Arabia. Some past studies [14], this study's limitation flows from its small sample, potential selection bias, cross-sectional design, and reliance on retrospective cost estimates. Longitudinal tracking of the expenses would be more productive in estimating the actual costs and projecting lifetime expenditures.

**Author Contributions:** Conceptualization, S.A.-A.; Methodology, S.A.-A. and A.A.; Software, S.A.-A. and A.A.; Validation, A.A. and K.A.; Formal analysis, K.A.; Investigation, S.A.-A., A.A. and K.A.; Resources, S.A.-A. and A.A.; Data curation, S.A.-A.; Writing—original draft, S.A.-A.; Writing—review & editing, K.A.; Visualization, S.A.-A. All authors have read and agreed to the published version of the manuscript.

**Funding:** This research received no external funding.

**Institutional Review Board Statement:** The study was conducted in accordance with the Declaration of Helsinki, and approval was obtained from the institutional review board of Al Baha University (protocol code 1443-21-43110073 and date of approval 21 March 2022).

**Informed Consent Statement:** Informed consent was obtained from all subjects involved in the study.

**Data Availability Statement:** The data presented in this study are available on request from the corresponding author.

**Conflicts of Interest:** The authors declare no conflict of interest.

## Appendix A. Result Excerpts from the Keratoconus Outcomes Research Questionnaire (KORQ)

**Table A1.** At what age were you diagnosed with keratoconus?

| | Diagnosis Age | No. of Respondents | Percent | Valid Percent | Cumulative Percent |
|---|---|---|---|---|---|
| | Less than 5 years | 1 | 1.1 | 1.1 | 1.1 |
| | 5–9 years | 2 | 2.2 | 2.2 | 3.4 |
| Valid | 10–14 years | 37 | 41.6 | 41.6 | 44.9 |
| | 15–19 years | 49 | 55.1 | 55.1 | 100.0 |
| | Total | 89 | 100.0 | 100.0 | |

**Table A2.** In which eye were you diagnosed with KC?

| | Which Eye | Frequency | Percent | Valid Percent | Cumulative Percent |
|---|---|---|---|---|---|
| | Left eye | 7 | 7.9 | 7.9 | 7.9 |
| | Right eye | 10 | 11.2 | 11.2 | 19.1 |
| Valid | Both eyes | 72 | 80.9 | 80.9 | 100.0 |
| | Total | 89 | 100.0 | 100.0 | |

**Table A3.** How long has it been since you were diagnosed with keratoconus.

| Disease Duration | | No. of Respondents | Percent | Valid Percent | Cumulative Percent |
|---|---|---|---|---|---|
| | Less than 5 | 24 | 27.0 | 27.0 | 27.0 |
| | 5–9 years | 23 | 25.8 | 25.8 | 52.8 |
| Valid | 10–14 years | 42 | 47.2 | 47.2 | 100.0 |
| | Total | 89 | 100.0 | 100.0 | |

**Table A4.** How many times have you had to buy glasses in the past year?

| | Buying Glasses | No. of Respondents | Percent | Valid Percent | Cumulative Percent |
|---|---|---|---|---|---|
| | Did not buy glasses last year | 33 | 37.1 | 37.1 | 37.1 |
| | Once in the past 12 months | 23 | 25.8 | 25.8 | 62.9 |
| Valid | Twice in the past 12 months | 20 | 22.5 | 22.5 | 85.4 |
| | More than thrice in the past 12 months | 13 | 14.6 | 14.6 | 100.0 |
| | Total | 89 | 100.0 | 100.0 | |

**Table A5.** Correlations.

| Keratoconus Duration | | How Long Has It Been since You Were Diagnosed with Keratoconus | Total Cost |
|---|---|---|---|
| How long has it been since you were diagnosed with keratoconus | Pearson Correlation | 1 | 0.214 * |
| | Sig. (2-tailed) | | 0.047 |
| | N | 89 | 87 |
| Total cost | Pearson Correlation | 0.214 * | 1 |
| | Sig. (2-tailed) | 0.047 | |
| | N | 87 | 87 |

* Correlation is significant at the 0.05 level (2-tailed).

**Appendix B. Keratoconus Outcomes Research Questionnaire (KORQ)**

1. Have you been diagnosed with keratoconus? *

   *Mark only one oval.*

   ( ) Yes

   ( ) No

   Demographic information

2. At what age were you diagnosed with keratoconus?

   ________________________

3. In which eye was keratoconus diagnosed?

   *Mark only one oval.*

   ( ) Right

   ( ) Left

   ( ) Both eyes

4.　Are you using glasses, contact lenses or other technology?

*Mark only one oval.*

◯ I use glasses

◯ I use soft contact lenses

◯ I use rigid contact lens (RGP)

◯ None

5.　Are you now, or have you ever received treatment for keratoconus?

*Mark only one oval.*

◯ Yes

◯ No

6.　What type of treatment did you receive?

*Mark only one oval.*

◯ Corneal Crosslinking

◯ Corneal Transplant Surgey

◯ Option 3

◯ Other: _________________________

7.　What is your vision (visual acuity) with glasses/contact lenses(If known only)

*Mark only one oval.*

◯ Left eye

◯ Right eye

8.   Your current glasses or contact lens prescription(If Known only)

*Mark only one oval.*

⬭ Left eye

⬭ Right eye

9.   Do you suffer from any other eye diseases? if so please specify

*Mark only one oval.*

⬭ No

⬭ Other: ___________________________

10.   Email *

_____________________________

11.   Address

_____________________________

_____________________________

_____________________________

_____________________________

_____________________________

12.   Phone number *

_____________________________

13. Gender

*Mark only one oval.*

◯ Male

◯ Female

14. Date of birth

_______________________________

*Example: January 7, 2019*

**Appendix C. Saudi Arabia Keratoconus Economic Burden Questionnaire**

Saudi Arabia Keratoconus Economic Burden Questionnaire                    https://docs.google.com/forms/u/0/d/1AKLsOBxgNzFXck7m_QIRdf5...

# Saudi Arabia Keratoconus Economic Burden Questionnaire

This form only seeks to gather information on the direct treatment and travel expenses that you incurred in relation to your keratoconus condition. it does not include any costs incurred on behalf of another person or treatment for a different condition.

\* Required

1. When were you diagnosed with keratoconus?

   _______________________________________

   *Example: January 7, 2019*

2. How many times have you had to buy glasses in the past year?

   *Mark only one oval.*

   ◯ I did NOT buy glasses last year

   ◯ 1 time

   ◯ 2 times

   ◯ More than 3 times

3. How frequently do you have to wear glasses or contact lenses in order to be able to see well?

   *Mark only one oval.*

   ◯ Many times every day

   ◯ Once or so in a week

   ◯ I almost never need glasses

   ◯ Never

4. How much out-of-pocket expenses did you incur to buy glasses or contact lenses over the last 12 months?

_______________________________

5. How much money do you spend to take care of your contact lenses and glasses (e.g., wipes)

_______________________________

Private Insurance Costs

> This section applies to expatriates and non-Saudis without access to Public Health Insurance.

6. Does your private insurance cover treatment related to keratoconus?

   *Mark only one oval.*

   ◯ Yes

   ◯ No

   ◯ I do not have private insurance

7. How much money do you pay as premiums for your private insurance cover per year (SAR)?

_______________________________

8. Are you happy with the rebates that you receive from your private insurance for treatment and other care related to your keratoconus condition? Please explain.

_______________________________

9. Do you think you would pay more in premiums to have an insurance policy that would give higher rebates for for glasses/contacts and other care related to keratoconus?

    *Mark only one oval.*

    ( ) Yes

    ( ) No

10. If you were not legally required to have private insurance, which of the following reasons explains why you would not buy a policy?

    *Mark only one oval.*

    ( ) I can manage without a private insurance cover

    ( ) It is too expensive

    ( ) It is bad value for money

    ( ) Other: ______________________________

11. In your opinion, which of the following expenses should be included in insurance rebates?

    *Mark only one oval.*

    ( ) Eye drops

    ( ) Glasses

    ( ) Surgical expenses

    ( ) Contact lenses fitting and solution

    ( ) Other: ______________________________

| Expenses Managing Keratoconus | This section covers any expenses that you have incurred in the past 12 months related to managing your keratoconus condition. |
| --- | --- |

12.  Did you receive care from an optometrist relating to your keratoconus condition over the apst 12 months?

*Mark only one oval.*

◯ Yes

◯ No

◯ Other: _______________________

13.  What clinic did you attend for the keratoconus care?

*Mark only one oval.*

◯ Goverment clinic or facility

◯ Private clinic or facility

◯ Not-for-profit clinic

14.  What services did you receive from these clinics?

*Mark only one oval.*

◯ Prescription for glasses

◯ Lens design and fitting

◯ General check-up

◯ Follow-up after a past appointment

◯ I was referred to the facility

◯ Other: _______________________

15.  How much did you incur in transport costs to visit a clinic for your keratoconus condition over the past 12 months (in SAR)?

_______________________________

16.   What other treatment expenses did you incur for your keratoconus condition
      (do not include any costs buying glasses, contact lenses and lens solutions)
      including the costs for consultations, check-ups, testing, lens fitting,
      hospitalisation and surgical fees?

17.   How much, in out-of-pocket expenses, did you incur on your treatment over the
      past 12 months?

18.   Other than the optometrists, did you receive keratoconus treatment or care
      from any other practitioners, specialists and/or hospital?

      *Mark only one oval.*

      ◯ Yes

      ◯ No

19.   What sort of eye clinics do you attend for treatment or care related to
      keratoconus?

      *Mark only one oval.*

      ◯ Public

      ◯ Private or not-for-profit

      ◯ I do not receive any additional care other than my optometrist

20. What was the nature of the care that you received from these practitioners or hospitals (other than optometrists)

_______________________________________

_______________________________________

_______________________________________

_______________________________________

_______________________________________

21. I was experiencing problems with my contact lenses

    *Mark only one oval.*

    ⬭ I needed to fit contact lenses

    ⬭ Follow-up on a past appointment

    ⬭ Routine check-up

    ⬭ Prescription check for my glasses

    ⬭ Surgery

    ⬭ Other: _______________________

22. Have you undergone any surgery to treat keratoconus?

    *Mark only one oval.*

    ⬭ Yes

    ⬭ No

23. If you underwent surgery to treat keratonus, what type o surgery was it?

    _______________________________________

24.    How much, in out-of-pocket expenses, did you incur on transport and
       accomodation realted to the said surgery?

       _______________________________________

25.    How much, in out-of-pocket expenses, did you incur for the actual surgical
       consultations, prep, procedure and post-operative care (in SAR)? This excludes
       the cost of glasses, contact lenses and lens solutions.

       _______________________________________

26.    Over the past twelve months, did you undergo therapy, support and/or other
       secondary care in relation to keratoconus?

       *Mark only one oval.*

       ( ) Yes

       ( ) No

27.    If Yes, what professionals did you receive care from?

       *Mark only one oval.*

       ( ) GP

       ( ) Psychologist

       ( ) Psychiatrist

       ( ) Homeopath

       ( ) Traditional Chinese Medicine

       ( ) Masseur

       ( ) Social support services

       ( ) Personal care services

       ( ) Other: _______________________

28. How much, in out-of-pocket expenses, did you spend on transport and accomoodation for these visits?

_________________________________

29. How much in, out-of-pocket expenses, did you spend for these services?

_________________________________

Informal Care and Support Costs

30. What do you do for a living?

*Mark only one oval.*

◯ Employed for wages

◯ Self-employed

◯ A homemaker

◯ A student

◯ Retired

◯ Unable to work because of keratoconus

◯ Other: _______________________

31. Have you ever had to change your career, job, leisure activity and/or course of study because of keratonus? *

*Mark only one oval.*

◯ Yes

◯ No

32. If you answered YES above, please explain why you needed to change

_________________________________

33. How much do you, your spouse and any other persons living with you earn in the past one year?

_______________________________

34. In the past 12 months, did you have to take time off work or otherwise was unable to work because of your keratoconus or because you had to receive treatment/care for keratoconus?

*Mark only one oval.*

◯ Yes

◯ No

35. How much money, do you think you lost as a result of your inability to work or need to take time off to receive treatment/care (in SAR)?

_______________________________

36. Over the past 12 months, were there times that you were unable to care for yourself or otherwise needed a helper to care for you because of keratoconus?

*Mark only one oval.*

◯ Yes

◯ No

37. How much did you pay or spend for the carer or assistive technology (including tips, wages, and transport, etc.)

_______________________________

38.   How much did you spend on the following medications over the apst 12 months?

*Mark only one oval.*

⬭  Prescription medicines, tablets, eye drops, etc.

⬭  keratoconus? etc.

⬭  Products: e.g. Low vision device, magnifier, cane,

⬭  Equipment: e.g. Special television or computer

⬭  screen, special computer software and telephone modifications

⬭  Other: _______________________________

39.   Please provide a list of the medications, equipment, and other items that you bought over the past 12 months. Where possible, provide the cost.

_______________________________________________

_______________________________________________

_______________________________________________

_______________________________________________

_______________________________________________

This content is neither created nor endorsed by Google.

Google Forms

40. How much did you spend on the following medications over the apst 12 months?

*Mark only one oval.*

- ◯ Prescription medicines, tablets, eye drops, etc.
- ◯ keratoconus? etc.
- ◯ Products: e.g. Low vision device, magnifier, cane,
- ◯ Equipment: e.g. Special television or computer
- ◯ screen, special computer software and telephone modifications
- ◯ Other: _______________________

41. Please provide a list of the medications, equipment, and other items that you bought over the past 12 months. Where possible, provide the cost.

_______________________

_______________________

_______________________

_______________________

_______________________

This content is neither created nor endorsed by Google.

Google Forms

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
