# Peer review of "The Lifetime Expenditure in People with Keratoconus in Saudi Arabia"

_2411-5150, 2023_

Round 1

Reviewer 1 Report

The Lifetime Expenditure In People With Keratoconus In Saudi Arabia

Line 8. It reads: “Results: The mean age and annual income of participants were 33.24 years and Saudi Riyal (SAR) 33,505.6180 (SD=62,215.29), respectively, with only 37% being employed for wages.”

Comment

Include the mean+/- standard deviation of the age.

In order to facilitate the understanding of the readers, the values of the amounts of money should be written in US dollars.

Line 28. It reads: “Keratoconus is an ectatic disorder characterised by progressive thinning, scarring, and anteriorly protrusion of the cornea, which results in irregular astigmatism, opacity, and impaired vision [1–3].”

Comment

Keratoconus involves protrusion of both anterior and posterior surfaces of the cornea.

In addition, since not always corneal opacity or scars appears, and sometimes corneal transplantation is required, it is important to mention these features. In addition, since crosslinking has changed the prognosis in keratoconus, it is important to mention it in the Introduction.

Consider modifying to:

“Keratoconus is an ectatic disorder characterised by progressive thinning, and protrusion of the cornea, which results in irregular astigmatism, and impaired vision. In advanced cases scarring and opacity may occur, and corneal transplantation can be required [1–3]. UV light-mediated corneal collagen crosslinking has been used for the last two decades, and has been shown to be effective in slowing the progression of the disease.”

Suggested additional references:

-Galvis V, Tello A, Carreño NI, Ortiz AI, Barrera R, Rodriguez CJ, Ochoa ME. Corneal Cross-Linking (with a Partial Deepithelization) in Keratoconus with Five Years of Follow-Up. Ophthalmol Eye Dis. 2016 May 12;8:17-21. doi: 10.4137/OED.S38364. PMID: 27199574; PMCID: PMC4869599.

-Raiskup F, Herber R, Lenk J, et al. Corneal Crosslinking With Riboflavin and UVA Light in Progressive Keratoconus: Fifteen-Year Results [published online ahead of print, 2023 Feb 1]. Am J Ophthalmol. 2023;250:95-102. doi:10.1016/j.ajo.2023.01.022

-Galvis V, Tello A, Laiton AN, Salcedo SLL. Indications and techniques of corneal transplantation in a referral center in Colombia, South America (2012-2016). Int Ophthalmol. 2019 Aug;39(8):1723-1733. doi: 10.1007/s10792-018-0994-z. Epub 2018 Jul 25. PMID: 30047076.

Line 30. It reads: “While its causation remains unknown, it was initially thought to affect 1 in every 2000 people[2,3]. With the advancements in diagnostic technologies, however, the incidence rates are now known to be more acute [2].”

Comment

These sentences are rather confusing.

Consider modifying to:

“While exact etiopathogenesis of KC remains elusive, both hereditary and environmental factors have been associated with the condition, and although for many years it was considered a non-inflammatory condition, for a couple of decades a growing body of evidence has been accumulating that suggests the participation of inflammatory pathways in the onset and progression of the disease. The prevalence of KC prevalence is highly variable in various geographical areas, and also with the greater availability in recent decades of corneal topography devices, diagnostic sensitivity has increased. Classically, based on a North American study, the prevalence was considered to be about 1 in 2000, but rates as high as 1 in 25 people have been reported in Southeast Asian populations. [2].”

Additional references:

-Galvis V, Tello A, Barrera R, Niño CA. Inflammation in Keratoconus. Cornea. 2015;34(8):e22-e23. doi:10.1097/ICO.0000000000000499.

-Rabinowitz YS, Galvis V, Tello A, Rueda D, García JD. Genetics vs chronic corneal mechanical trauma in the etiology of keratoconus. Exp Eye Res. 2021;202:108328. doi:10.1016/j.exer.2020.108328

 Line 36. It reads: “Ordinarily, the cornea is elliptical. It steepens gently towards the central corneal zone and it nearly perfectly flattens between the intermediate and the peripheral corneal zones, such that its curvature’s radius varies evenly from the centre towards the periphery. In patients with keratoconus, the corneal apex often occurs in the lower region and is severely protruded. This results in an uneven corneal shape [2].”

Comment

These sentences are not really an accurate description of the keratoconus.

Consider modifying to:

 “Usually the normal cornea is prolate, i.e. more curve in the center, and its main meridians are orthogonal. In patients with keratoconus, the corneal apex bulges eccentrically, more frequently in the inferotemporal quadrant, causing irregular astigmatism[2].”

Line 41. It reads: “As the disorder is progressive, the corneal shape and extent of astigmatism are usually mild at the onset, which is why early-stage keratoconus is correctable with either soft contact lenses or glasses. While rigid gas-permeable hard contact lenses are contra-indicated, spherical hard contact lenses may be used as they have an even structure with an evenly reducing curvature radius towards the periphery [2]. The characteristically uneven corneal shape renders the use of hard contact lenses impractical as the disease advances, but aspherical and multi-curve hard contact lenses may still be used [1,2].”

Comment

The wording of the first sentence could be improved.

In addition, the second sentence is confusing and inaccurate. Firstly, there is no consensus about rigid gas-permeable contact lenses and keratoconus, and actually, they are the most frequent non-surgical treatment alternative for patients with clinically significant keratoconus. Whether the rigid gas-permeable contact lenses can hasten the progression of the disease  is an unsettled topic, and further studies are warranted.

In addition, the aspherical (not “spherical” as written in the text) and multi-curve contact lenses whose results were published in the cited article (Kumanomido et al. 2022) are types of rigid gas-permeable hard contact lenses, therefore the last sentence can be incorporated into the second.

Therefore, consider changing the text to:

“As the disorder is usually slowly progressive, the corneal shape and extent of astigmatism are typically mild at the onset, which is why early-stage keratoconus is correctable with either glasses or soft contact lenses. Rigid gas-permeable hard contact lenses, including special aspheric and multi-curve designs (which are preferable in advanced cases with a more uneven corneal surface), are the most frequent non-surgical treatment alternative currently used worldwide for patients with clinically significant keratoconus. However, whether the rigid gas-permeable contact lenses can hasten the progression of the disease is an unsettled topic, and further studies are warranted. Scleral lenses, which vault over the limbus and cornea without having contact with them, on the other hand, are more expensive and difficult to fit, but they can provide good vision even in very irregular corneas, and their use is increasing. [2]. Eventually, however, a percentage of eyes with advanced keratoconus and intolerance to contact lenses or poor visual acuity even with these optical devices will require a corneal transplant.”

Additional References:

-Zhang XH, Li X. Effect of rigid gas permeable contact lens on keratoconus progression: a review. Int J Ophthalmol. 2020 Jul 18;13(7):1124-1131. doi: 10.18240/ijo.2020.07.17. PMID: 32685402; PMCID: PMC7321943.

-Koppen C, Kreps EO, Anthonissen L, Van Hoey M, Dhubhghaill SN, Vermeulen L. Scleral Lenses Reduce the Need for Corneal Transplants in Severe Keratoconus. Am J Ophthalmol. 2018 Jan;185:43-47. doi: 10.1016/j.ajo.2017.10.022. Epub 2017 Nov 16. PMID: 29103959.

-Galvis V, Tello A, Laiton AN, Salcedo SLL. Indications and techniques of corneal transplantation in a referral center in Colombia, South America (2012-2016). Int Ophthalmol. 2019 Aug;39(8):1723-1733. doi: 10.1007/s10792-018-0994-z. Epub 2018 Jul 25. PMID: 30047076.

Line 49. It reads: “The extant empirical evidence shows that the prevalence of keratoconus in Saudi Arabia is comparably higher than in other countries, possibly because of geographical/regional, environmental, and genetic differences, as well as differences in diagnostic technologies”.

Comment

The adjective “extant” (meaning “still existing; not destroyed”) does not really fit the sentence, and in reality, it is superfluous in the context. In addition, since in Europe, North America, and many other countries of Asia and Latin America, modern corneal diagnostic technologies are available,  and the rate of keratoconus seems to be much lower, differences in diagnostic technologies are not possibly a factor in such differences.   

Consider modifying to: “The empirical evidence shows that the prevalence of keratoconus in Saudi Arabia is comparably higher than in other countries, possibly because of geographical/regional, environmental, and genetic differences”.

Line 53. It reads: “On their part, Althomali et al. [6] screened a sample of 687 patients (353 females) that had undergone routine pre-operative evaluation at a facility in Taif in 2014-2015. They found the prevalence of manifest keratoconus at 8.59%, with 6.55% and 2.04% having bilateral manifest keratoconus and unilateral manifest keratoconus, respectively. Further, the study found sub-clinical bilateral and unilateral keratoconus in 9.46% and 6.55% of the sample, respectively [6].”

Comment

The study by Althomali et al. was performed among patients seeking laser vision correction, therefore is a self-selected population, and therefore suffers from selection bias, but it still could be compared to studies also performed on refractive surgery candidates in  Saudi Arabia and other countries. In fact, Al-Amri found a much higher rate in Abha (Saudi Arabia) among patients interested in undergoing refractive surgery (24% had keratoconus and 17.5% were keratoconus suspects). Since the idea is corroborate that Saudi Arabia has a high prevalence of keratoconus, consider modifying it to: “Two studies among individuals seeking refractive surgery have shown in Saudi Arabia prevalence of manifest keratoconus between 8.6% and 24 %. Although due to selection bias, these studies could show higher prevalence than in the general population,  is possible to compare them with similar studies in other countries, and these rates were higher than those found in refractive surgery services in Central Europe,  North America, and South America [6].”

Additional references:

-Al-Amri AM. Prevalence of Keratoconus in a Refractive Surgery Population. J Ophthalmol. 2018;2018:5983530. Published 2018 Sep 6. doi:10.1155/2018/5983530

-Galvis V, Tello A, Jaramillo JA, Gutierrez AJ, Rodriguez L, Quintero MP. Prevalence of keratoconus patients who consulted with a desire refractive surgery in ophthalmology center reference Bucaramanga, Colombia. Rev Soc Colomb Oft al. 2011;44:129-134.

-Nesburn AB, Bahri S, Salz J, Rabinowitz YS, Maguen E, Hofbauer J, Berlin M, Macy JI (1995) Keratoconus detected by videokeratography in candidates for photorefractive

keratectomy. J Refract Surg 11(3):194–201.

-Gilevska F, Kostovska B, Osmani I, et al. Prevalence of keratoconus in refractive surgery practice population in North Macedonia. Int Ophthalmol. 2022;42(10):3191-3198. doi:10.1007/s10792-022-02319-0.

Author Response

  1. Mean+/- standard deviation of the age added at line 8 (blue font). Monetary values have been converted to USD throughout the document.
  2. As suggested, the following was inserted at line 28-39:

“Keratoconus is an ectatic disorder characterised by progressive thinning, and protrusion of the cornea, which results in irregular astigmatism, and impaired vision. In advanced cases scarring and opacity may occur, and corneal transplantation can be required [1–3]. UV light-mediated corneal collagen crosslinking has been used for the last two decades, and has been shown to be effective in slowing the progression of the disease.”

  1. Line 30 (now at 40) has been edited for clarity and the suggested text has been adopted in full (changes in blue font at lines 40 onwards) reading

 “While exact etiopathogenesis of KC remains elusive, both hereditary and environmental factors have been associated with the condition, and although for many years it was considered a non-inflammatory condition, for a couple of decades a growing body of evidence has been accumulating that suggests the participation of inflammatory pathways in the onset and progression of the disease.”

  1. Line 36 (now 46) modified as suggested (line 47-49).
  2. Line 41 (now 50) modified for clarity and accuracy as suggested. Referenced added.
  3. Line 49 (now at 64) modified as suggested with “extant” expunged.
  4. Line 53 (now at 68) modified as suggested with the insertion of the following with citation:

“Two studies among individuals seeking refractive surgery have shown in Saudi Arabia prevalence of manifest keratoconus between 8.6% and 24 %. Although due to selection bias, these studies could show higher prevalence than in the general population,  is possible to compare them with similar studies in other countries, and these rates were higher than those found in refractive surgery services in Central Europe,  North America, and South America [13,14].”

Reviewer 2 Report

Dear Authors,

Thank You for presenting Your manusctipt: "The Lifetime Expenditure In People With Keratoconus In Saudi Arabia". It tties to describe very importiant, often underestimated and overlooked proble of socioeconomic burden. This cross-sectional  study was conducted using online survey. 

There are some minor problems with the manuscript:

Section introduction describes nature of the disease. However it considers only refractive correction and does not discuss other treatment options (cross-linking, keratoplasty). In line 32 word "acute" in my opinion is not the best choice. 

In section methodology please list other statistical data analysis applications which You mentioned.

For better understanding socioeconomicl context please give actual for Your data Saudi Riyal to dollar exchange rate.

In tables word "Frequency" for columns is in my opinion not proper - should be number or n.

In figure 2 cross-linking is not shown. Please write scleral contact lens - or slceral CL for better clarity. 

In appendix there are two questionares: the first one is not named, what creates ambiguities. 

The one of the main problem is short observation period (12 month). As the keratoconus could be stable in older group of patients, it seems to be obvious that the need of changing glasses could be low (37% did not buy glasses).  The correlation from figure 8 is not explained clearly- The treatment costs increased with disease duration - why, is the correlation linar or it reaches stailisation? 

The major drawback of the manuscript is lack of correaltion of visual acuity and stage of the disease with socioeconomic situation.

According to results: undergoing surgery to correct keratoconus has (not significant) a negative effect on the lifestyle and medical costs of keratoconus -- please discuss, if surgical treatment make sense.

The including criteria are not given, espetially if keratoconus coexists with any other diseases (atopy, Down SYndrome e.g.) what could also infuuence results.

I suggest  Reconsidering publishing the masnuscript after major revision.

Author Response

  1. Inserted the following sentences at 31-37 to discuss other treatment options:

“In early stages, keratoconus may be managed using standard spectacles or contact lenses, even though worsening astigmatism necessitates scleral or contact lenses. Highly progressive cases render it difficult to fit rigid contact lenses, by which time, corneal transplantation becomes necessary[4]. Other than Intacs®, penetrating keratoplasty is easily one of the most common surgical approach, even though the latter procedure is fraught with fairly high rates of post-operative complications, including secondary glaucoma, cataracts, and graft rejection[4–7].”

  1. In line 32 word "acute" has been expunged for accuracy. 
  2. In the methodology, the last sentence has been rephrased to “Appropriate descriptive and inferential statistics were computed by use of SPSS, Eviews and Microsoft Excel.”
  3. The monetary values throughout the document are now expressed in USD.
  4. The label “Frequency” in all Tables has been replaced with “No. of Respondents”.
  5. There was no response indicating cross-linking. Labels in Figure 2 have been modified to indicate “scleral contact lens” for clarity instead of “scleral” (page 5). 
  6. The confusion has been clarified. There is only one questionnaire, now labelled Appendix B. Appendix A contains some results that could not be included in the main body of the paper.
  7. The following has been inserted at 223 to explain the correlation coefficient:

“Even so, the linear correlation coefficient is 0.038, which is near negligible co-variation of the duration since diagnosis and the twelve-month expenditure. This outcome could be explained by the fact that twelve-month period of observation is relatively short since major treatment expenditures (such as surgery and change of glasses) could remain relatively constant within a given year.”

  1. In response to the comment “The major drawback of the manuscript is lack of correlation of visual acuity and stage of the disease with socioeconomic situation,” the correlation Table 7 has been re-inserted to include proxies for socio-economic status i.e., annual income and age as well as measures of visual acuity. Even so, none of these variables exhibit statistically significant correlations, except Age and total Cost.
  2. The statistically insignificant result in respect to surgery is discussed on page 15 as follows:

“The results are inconclusive on the cost impact of undergoing keratoconus surgery, attending either public or private clinics, and assisted living due to keratoconus. Thus, more research, with larger and more robust sampling is required to settle these findings. It is notable that none of the respondents indicated having undergone corneal cross-linking, which is the only treatment (approved by the US Food and Drug Administration) that has been shown to stem the progression of keratoconus. The effectiveness of surgery could be factor in the results, as penetrating keratoplasty struggles with high rates of post-operative complications[4–7]. ”

  1. Exclusion criterion is included under sub-section 3.2:

“Respondents who suffer from acute, chronic or genetic/congenital non-visual comorbid conditions such as Downs Syndrome, Marfan syndrome, GAPO syndrome, osteogenesis imperfecta, and Ehlers–Danlos syndrome were excluded.”

Round 2

Reviewer 1 Report

Interesting study

Author Response

Dear reviewer 

Thank you for your effort in reviewing the process of this manuscript

Reviewer 2 Report

Dear Authors,

Thank You for improvement of Your article, which gives readers good insight in expediture of keratokonus in Saudi Arabia, a country with very high prevelence of the disease. The article analises  the socioeconomic burdens and costs, showing iits corelations with visual aquity, age, age of diagnosis and durationon of the disease.

The data are clear presented, but figure 8 must be improved (costs must be closed from both sides).

Despite specific economical situation in countries and regions due to insurence systems and medical care organisation, the article could give readers overview of financial burden of keratoconus, which are in some aspeksts universal and could be extrapolated on other contries.

Therefore I accept the article for publication (after small correction in figure 8).

Author Response

Dear reviewer 

Thank you for your effort in reviewing the process of this manuscript.

Figure no 8 has been improved and I hope that's what you mean

Thank you

Round 3

Reviewer 2 Report

Dear Authors,

thank You for improving Your manuscript. However the table 8 has still the same mistake: the ranges of values paid as premiums for your private insurance cover per year must be more prcise: 0,  from 0 to 945.  from 945 to 1350, from 1350 to 2700, >2700, otherwise it could be misunderstood by the readers. Please correct it, or axplain me Your opinion and point of view. Friendly regards.

Author Response

Dear reviewer

the comment about Figure 8 has been addressed now

Thank you
